# Does the perception of red tape affect the emotional labor of frontline retail staff in China? A post-COVID-19 era

Qin Qiang[1]¤, Kaixin Wang[2], Jianxin Lai[3]*

1 School of Journalism and Communication, Renmin University of China, Beijing, China, 2 School of Public Administration, Beijing University of Aeronautics and Astronautics, Beijing, China, 3 School of policy and management, Tsinghua University, Beijing, China

¤ Current Address: School of Journalism and Communication, Renmin University of China,Beijing,China
* qinqiang@ruc.edu.cn

## Abstract

Red tape denotes valid rules or procedures that contribute nothing to achieving an organization's goals. How red tape affects corporate employees' emotional labor has received little attention. Therefore, this study recruited retail staff from Beijing after the COVID-19 pandemic, assessing the roles of negative emotions and customer orientation. Red tape negatively affected surface and deep acting among retail employees. Negative emotions mediated the relationships between red tape and surface and deep acting. Customer orientation moderated the relationship between negative emotions and surface acting as well as deep acting. Lastly, implications are provided for companies to promote employee emotional labor.While these measures were operationally necessary, their prolonged enforcement and administrative burden led employees to perceive them as bureaucratic obstacles rather than supportive regulations. Employees with high customer orientation may adopt alternative regulation strategies, such as cognitive reappraisal, instead of surface acting.

## Introduction

After the coronavirus 2019 (COVID-19) pandemic, the global service industry has been severely impacted, especially in China, where pandemic control measures have been relatively not strict. The post – pandemic period is significantly different from the pre – pandemic period in many aspects. During the pandemic, retail businesses faced strict 防控 measures, which led to a large number of new workplace rules. These rules not only changed the daily operations of retail stores but also had a profound impact on employees' work experience. After the pandemic, as the situation gradually returned to normal, the remnants of these rules, which we classify as red tape, still influenced employees' emotional labor. Compared to the pre – pandemic period, employees now have to deal with the legacy of these rules while also

**Data availability statement:** All relevant data are within the paper and its Supporting Information files.

**Funding:** The author(s) received no specific funding for this work.

**Competing interests:** The authors have declared that no competing interests exist.

adapting to the changing market demand. For example, the shift in customer behavior and expectations after the pandemic requires employees to adjust their service strategies, but the red tape may limit their ability to do so effectively. Therefore, focusing on the post – pandemic period allows us to explore how these new factors interact and affect employees' emotional labor, providing practical implications for businesses to better manage their employees in the new normal.As the pandemic control measures gradually relaxed, the physical business sectors began to recover and again began providing various customer services. During this process, frontline retail staffs' service quality and emotional labor have become particularly important, as they directly affect customer satisfaction, corporate reputation, and social stability [1]. "Emotional labor" refers to employees regulating or managing their emotions to achieve a public-facing facial or bodily appearance [2]. In existing research, antecedent variables of emotional labor in the retail industry include several levels: at the individual level, there are influences of personality trait tendencies [3], emotional resources [4], and service motivation [4]; at the organizational level, there are effects of organizational support [5], organizational emotional expression [6], and job characteristics [7].

However, a crucial factor influencing the emotional labor of retail employees during the pandemic has been neglected in previous studies. Frontline service workers in the retail industry endured the impacts of the Covid-19 pandemic, facing a multitude of red – tape – related situations. For example, they had to carry out repeated customer entry registrations, monitor customers to ensure mask – wearing compliance, participate in frequent but unproductive online meetings, and submit numerous work summaries. This was particularly the case in countries that implemented strict control measures to prevent the resurgence of the virus.

Red tape promulgated by government administrations has been attracting increasing attention in the field of public management, which is considered the most typical bureaucratic structural feature that affects individual behavior in the public sector [8]. It not only weakens organizational performance and reduces government work efficiency at the organizational level, but also increases organizational employees' propensity to leave, reduces organizational commitment, reduces employee output [9], affects employees' active work behavior [10,11], and even affects government-citizen interactions. Hence, red tape is typically perceived as a bureaucratic disease and widely criticized. While red tape is often closely associated with government bureaucratic systems, where unnecessary and overly elaborate formalities are more prominent, employees of businesses, especially large companies, also experience red tape, such as filling out numerous reports, adhering to excessive rules, and receiving frequent supervision [9,12]. By comparing public and private organizations, researchers have found that businesses may involve more red tape than the government [13], and this can induce negative emotions in employees [14], thereby having a significant impact on the emotional labor of frontline retail staff in the service industry. Given that there is currently little literature focusing on the impact of red tape on the emotional labor of frontline retail staff, this study addressed how organizational red tape impacts employees' emotional labor. Based on the conservation of resources theory (COR)

theory and the job demands-resources (JD-R) model, and data from physical retail store staff obtained through question-naires, we outlined the impact of organizational red tape on employees' emotional labor and explored the roles of negative emotions and customer orientation in this relationship. This study provides effective suggestions for organizations to improve employees' emotional labor and promote organizational development.

According to the COR theory and the JD-R model, the red tape may lead employees to experience negative emotions, such as frustration, dissatisfaction, or stress [15]. Frontline retail employees, as service providers, consider customer service as the nature of their job, and employees with a high customer orientation may regulate their emotions [16]. Therefore, this study introduced negative emotions as a mediating variable and customer orientation as a moderating variable of the relationship between red tape and the emotional labor of retail employees.

Our study makes the following contributions. First, we help illuminate the antecedents of emotional labor. Previous studies explored the impact of individual characteristics, emotional resources, service motivation, and organizational support on emotional labor. We introduced red tape, which is often discussed in public organizations, and focused on the impact of high red tape situations during the pandemic on the emotional labor of frontline retail workers. Second, concerning the public administration literature, we incorporated the impact of red tape on emotional labor into the COR and JD-R models. Red tape was considered a resource drain and job demand for employees; thus, testing whether and how it affected employees' emotional work provided insights into its practical significance in business. Third, we verified the importance of negative emotions and customer orientation in the impact of red tape on employees' emotional labor, which helped us better understand the relationship between red tape and emotional labor. Finally, this study enriches the existing literature on red tape. Past studies involving red tape have often focused on comparing red tape in public and private organizations and the negative impact of red tape in public organizations. This study extends this knowledge to frontline retail service personnel, improving our understanding of the impact of red tape.

## Conceptual framework and hypotheses

### COR theory and JDR model

COR theory and the JD-R model both explain how resources in the work environment affect an individual's work. According to the COR theory, employees tend to strive to acquire and retain resources and do their best to avoid risks that can lead to resource loss [17]. Brotheridge and Lee (2002) suggested that emotional labor consumes emotional and psychological resources [16]. To minimize this consumption, individuals within an organization tend to retain valuable emotional resources to achieve a balance. The impact of emotional resources on individuals' choice of emotional expression methods in an organization follows two main paths. In the relationship between emotional regulation ability and emotional labor strategies, employees with stronger emotional regulation capabilities are better at managing their emotions. According to the COR theory, they can better balance the consumption and gain of emotional resources. When faced with job demands, these employees are more likely to choose deep – acting strategies because they can effectively regulate their emotions to meet the organization's emotional expression rules without excessive depletion of emotional resources. In contrast, employees with weaker emotional regulation abilities may find it more difficult to engage in deep acting and are more likely to rely on surface – acting strategies, as surface acting requires less internal emotional adjustment. Although this relationship is not explicitly tested in our statistical models, we now incorporate it visually in the theoretical framework to illustrate how emotional regulation ability may influence the choice of emotional labor strategies through resource preservation logic It is an important underlying mechanism that influences the choice of emotional labor strategies and should be considered in future research.First, individuals who experience negative emotions can experience a loss of emotional resources, causing them to assign more importance to the potential risk of resource loss than to the acquisition of resources. To avoid the loss of emotional resources, employees who are experiencing negative emotions tend to choose physiological responses to express emotions, such as body expressions and professional rigid smiles, which are surface acting, rather than regulating

emotions subjectively, which is deep acting. Second, employees with good emotion regulation abilities retain more emotional resources. These people value the acquisition of new emotional resources and pay less attention to the loss of emotional resources [18]. Therefore, employees with better emotion regulation abilities tend to choose deep-acting strategies. Deep acting reflects better service quality in customer service [4], which in turn increases the emotional resources of employees. It's important to note that while deep acting can result in emotional resource gains, it is also resource – intensive. This means that employees need to invest a significant amount of mental and emotional energy to genuinely align their internal emotions with the organization's requirements. This duality of deep acting may explain the inconsistencies found in prior research. For example, some studies might have observed positive outcomes associated with deep acting, such as enhanced customer satisfaction and employee well – being, while others might have found that the high resource consumption of deep acting led to negative consequences like increased emotional exhaustion. Acknowledging this duality provides a more comprehensive understanding of the role of deep acting in the context of emotional labor.

The JD-R model is a specific application of COR theory that is used to explain the significant role of resources in the work environment for individuals [19]. Job demands-resources (JD-R) theory suggests that job characteristics can be divided into job demands and job resources [20]. Correspondingly, work can affect employees through two pathways: health impairment and motivation gain [21]. When job demands are too high or resources are insufficient, a process of resource and energy depletion is triggered, leading to negative organizational outcomes (such as poor job performance) [22]. Meanwhile, abundant job resources can stimulate employee motivation gains, thus producing positive work impacts (such as high organizational commitment) [23]. However, when the JD-R model was proposed, it did not consider specific occupations and situations [8]. Therefore, we introduced red tape and emotional labor into the JD-R model.

This study, based on COR theory and the JD-R model, examined the impact of organizational red tape on employees' emotional labor. When work is cumbersome or demands are high, emotional and resource depletion can occur, subsequently affecting emotional labor. Therefore, this study introduced negative emotion as a mediating variable to verify the relationship between red tape and emotional labor. Employees with better emotion regulation capabilities often retain more emotional resources. Hence, the concept of customer orientation was introduced. When employees' customer orientation is high, they can regulate their emotional resource depletion.

## Red tape and emotional labor

The term "red tape" originated from 19th-century British public administration practices, in which official government documents were traditionally tied with red tape. Over time, people gradually associated red tape with bureaucracy and imbued it with symbolic significance, describing the proliferation of government documents, cumbersome rules, and the resulting procrastination and delays [24]. In the National Performance Review's 1993 report, "From Red Tape to Results: Creating a Government That Works Better and Costs Less," academia began to pay attention to the phenomenon of red tape. Bozeman(1993) provided a relatively clear and classic definition of red tape: it is a phenomenon in which cumbersome procedures and excessive rules in an administrative system led to inefficient government operations [24]. It stems from inherently bad rules or formalized rule systems that lose their intended functions and transform into cumbersome and useless rule systems. "During the COVID - 19 pandemic, some of the workplace rules, such as repeated customer entry registration, frequent supervision of customers wearing masks, and numerous work summaries, were initially implemented to prevent the spread of the virus. However, over time, in the context of retail work, these rules became overly burdensome and inefficient. For example, some retail stores found that the repeated entry registration process caused long queues at the entrance, which not only affected customer experience but also consumed a large amount of employees' time and energy. And the frequent online meetings and work summaries often focused on redundant information, without effectively contributing to improving service quality or business performance. From the perspective of employees, these rules became a form of red tape. Although they had an operational necessity during the early stage of the pandemic, as the situation changed, their negative impact on employees' work efficiency and emotional labor became more prominent.

 

Therefore, we classify these rules as red tape in the context of our research to explore their impact on employees' emotional labor.For example, "useless documents," "unnecessary paperwork" "overzealous recordkeeping" and "frequent inspections" [10] are manifestations of red tape. This concept has gradually become a widely accepted definition. Bozeman later expanded upon his research, suggesting that organizational members' perceptions of organizational red tape vary and that there is a perceptual threshold for red tape. That is, red tape is only considered onerous when an individual or organization feels that the degree of red tape exceeds its expected acceptable level [25]. This study viewed red tape as a subjective judgment, perceivable by the organization and its members, and referring to the recognition of rules and procedures within the organization that must be adhered to despite being redundant or meaningless. While the concept of red tape originates from public administration, recent studies have demonstrated its applicability in business settings, especially in large chain stores where internal procedures become increasingly bureaucratic. In contrast, small retailers may experience less systemic red tape but may still suffer from informal constraints or customer-facing inefficiencies. This contrast deserves further empirical exploration in future research.

With the rapid development of the service industry, service quality has become a core factor in competitiveness in the market. Correspondingly, emotional labor, which is closely related to the service quality of the service industry, has received increasing attention [18,26]. Currently, emotional labor has been typically divided into two dimensions according to display strategies: surface acting and deep acting. Surface acting refers to situations in which employees' internal emotional perceptions do not align with the emotional expression rules set by their organization. During the emotional expression process, employees control and regulate their surface expressions such as facial expressions, body movements, and tone of voice to display behaviors that are consistent with the organization's rules [27]. Hence, surface acting is also viewed as "acting in bad faith" [28]. In contrast to surface acting, deep acting refers to instances in which employees' true inner emotions do match the organization's emotional expression rules. Employees not only ensure that their specific emotional expressions align with the organization's rules but also genuinely present the emotions that the organization requires at a deeper, subjective level. Thus, deep acting is also known as "acting in good faith" [28].

COR theory suggests that when employees face resource loss, they experience stress. During the pandemic, frontline personnel in the retail industry were required to scan codes, disinfect at all times, register customers, and keep customers at a distance. These tasks can lead to the depletion and loss of employees' psychological and physical resources. This resource loss might result in employees feeling stressed, leaving them without the energy to invest in emotional labor [29]. At the same time, JD-R theory posits that when job demands are excessive and the norms to be adhered to are overly complex, employees' energy is drained, affecting their emotional labor. Although some established research on emotional dissonance and surface acting indicates that emotional dissonance often leads to surface acting, our hypotheses are based on a different perspective. In the context of red tape, employees may experience a unique stressor. According to the COR theory, red tape – related resource depletion can cause employees to feel overwhelmed. For example, research by has shown that excessive administrative tasks similar to red tape can lead to a significant increase in employees' negative emotions [26]. These negative emotions may disrupt the normal emotional regulation process of employees. Instead of simply using surface acting as a response to emotional dissonance, employees may reduce surface acting because they lack the emotional resources to engage in even this form of emotional labor. This logic departs from traditional dissonance-based models, which assume employees always retain enough capacity for surface acting. In high red tape environments, employees may reach a "resource exhaustion threshold" where even minimal effort to feign emotions becomes too costly. Under COR theory, individuals facing continuous resource loss tend to withdraw from optional demands, including superficial displays of compliance. Hence, the decline in surface acting can be interpreted as a strategic disengagement from emotionally taxing behaviors, rather than a failure to comply. Empirical evidence also supports our hypotheses. In a recent study of [13], it was found that when employees were exposed to high levels of red tape – like situations, their surface acting decreased significantly along with an increase in negative emotions.Therefore, the following hypotheses were proposed:

**Hypothesis 1** Red tape negatively affects the surface-acting behavior of employees.
**Hypothesis 2** Red tape negatively affects the deep-acting behavior of employees.

## The mediating role of negative emotions

According to COR Theory, emotional resources affect emotional labor strategies [17]. Red tape depletes employees' emotional resources. Meanwhile, JD-R theory posits that excessive and complex job demands can make employees feel stress, anxiety, and tension, among other negative emotions [30]. Employees who are experiencing negative emotions tend to have fewer emotional resources. Consequently, they pay more attention to the loss of resources than to their acquisition. Reducing the loss of these emotional resources severely affects the emotional labor of employees who are experiencing negative emotions. Employees within an organization first need to integrate into the organization and are constrained by the organization's emotional expression rules [31]; second, they need to serve customers and make efforts to engage in emotional labor [32]. Finally, their own negative emotions can also consume emotional resources [33].

Therefore, employees are likely to be influenced by the organization's emotional expression rules and their experience of negative emotions, resulting in emotional labor. According to COR theory, if employees use emotional expression strategies to comply with the organization's emotional expression rules and already harbor negative emotions, their emotional labor will be negatively affected. There are researchers found that negative emotions have a significant negative impact on surface and deep-acting strategies [34]. Based on the above background, the following hypotheses were proposed:
**Hypothesis 3** Negative emotions mediate the relationship between red tape and surface acting. That is, red tape increases negative emotions, thereby inhibiting surface acting among employees.
**Hypothesis 4** Negative emotions mediate the relationship between red tape and deep acting. That is, red tape increases negative emotions, thereby inhibiting deep acting among employees.

## The moderating role of customer orientation

Customer orientation refers to "the employee's behaviors that are geared towards satisfying customers' needs adequately" [35], which is a key focus for any service organization. Customer orientation is a type of psychological resource that includes attitude, mindset and work ethics [34]. Customer orientation is primarily driven by the principle of "customer care", which shifts the attention of salespeople from their own benefits to the needs and satisfaction of their customers. Empirical studies suggest that employees who are customer-oriented can promote customer service perceptions, and display proactive behavior, which have a positive effect on organizations.

Customer Orientation can indeed help mitigate the negative emotions experienced by employees. This is primarily because a customer-oriented mindset encourages employees to view their roles from the perspective of customer satisfaction, which can result in a more positive work experience. Indeed, individuals with high customer orientation might experience stronger negative emotions resulting from red tape. According to the COR, individuals usually strive to protect their resources and prevent resource loss when faced with stress [17]. These resources can be time and money, as well as self-esteem and emotions. In the service industry, employees are required to invest substantial emotional resources to meet customer needs and expectations For those with a high degree of customer orientation, their psychological resources are primarily dedicated to satisfying customer demands and delivering high-quality service [36]. However, bureaucratic behaviors such as red tape can hinder their ability to provide excellent service, depleting their psychological resources and leading to negative emotions.

Red tape can trigger fears of resource loss and induce stress, particularly in highly customer-oriented employees [5]. They may feel frustrated because they are unable to utilize their resources – their skills, time, and energy – to achieve their goal of satisfying customer needs [36]. The discrepancy between their desire to provide excellent customer service and the restrictions imposed by bureaucracy can lead to emotional dissonance, a form of psychological strain [37].

However, employees with high customer orientation may not always resort to surface acting when red tape disrupts their service delivery. Instead, they may engage in adaptive emotion regulation strategies such as cognitive reappraisal—reframing frustrating procedures as manageable challenges or temporary obstacles. This reframing helps conserve emotional resources and sustain deep acting or authentic emotional expression, rather than relying on inauthentic surface displays. Therefore, the following hypotheses were proposed:

**Hypothesis 5** Customer orientation moderates the effect of negative emotions on surface acting, thus moderating the mediating effect of negative emotions. That is, higher customer orientation amplifies the impact of negative emotions caused by red tape on surface acting.

**Hypothesis 6** Customer orientation moderates the effect of negative emotions on deep acting, thus moderating the mediating effect of negative emotions. That is, higher customer orientation of employees amplifies the impact of negative emotions caused by red tape on deep acting. The theoratical framework see Fig 1.

## Research methodology

### Sample and procedures

Data were collected from frontline retail employees at Beijing malls from 20th/December/2023–15th/February/2024. We conducted our study by enlisting participants online and distributed digital surveys to them. Upon completion of the survey, participants were offered a nominal monetary compensation. To avoid common method bias, we measured in two separate practice stages, once for the independent variable (i.e., red tape) and the mediator (i.e., negative emotions), and once for the dependent variables (i.e., surface acting and deep acting) and the moderator (i.e., customer orientation). In our mediation analysis, we employed a time – lagged design to reduce the potential for reverse causality. However, we acknowledge that this design may not completely eliminate all biases. There could still be unobserved variables that influence both the independent variable (red tape), the mediator (negative emotions), and the dependent variables (surface acting and deep acting). For example, the overall work environment of the retail industry, which may include factors such as store culture and management style, could affect employees' perception of red tape, their emotional state, and their choice of emotional labor strategies. Although we have controlled for some demographic variables, these unobserved factors may still lead to biases in our results. Future research could consider using more advanced statistical methods, such as instrumental variable analysis, to further address these potential biases.

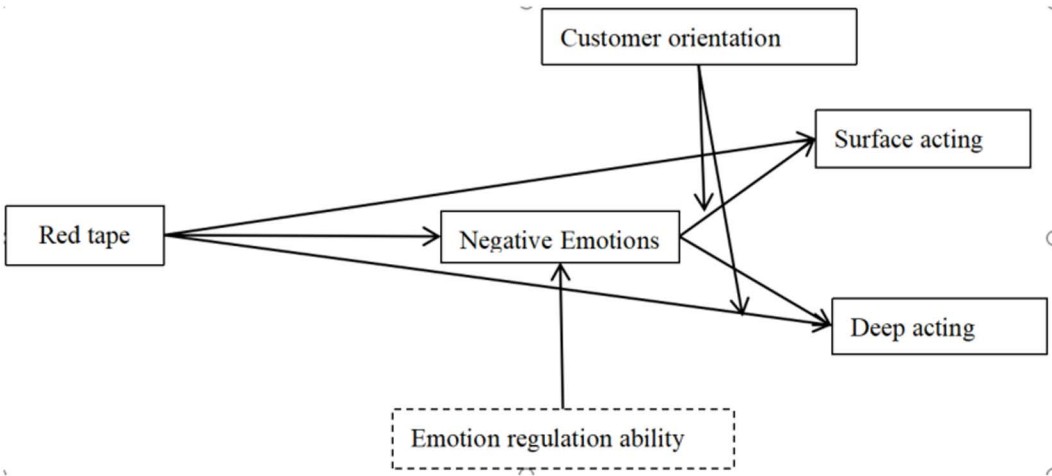

**Fig 1. Illustrates the theoretical and logical framework of this study.**

All procedures in this study involving human participants were conducted in strict accordance with the ethical standards of the Institutional Review Board,School of Journalism and Communication Renmin University of China.

The measurements reflected the overall situation of the past three months. The time interval between two waves was three weeks, with 523 responses at the first wave and 412 at the second wave. Any questionnaire containing incorrect responses was eliminated through a thorough review process, yielding 396 valid responses at last. Males accounted for 56.6% of the sample, whereas females accounted for 43.4%. Those under 30 years of age accounted for 61.1% of the total sample, 31–40 years old accounted for 30.1%, 41–50 years old accounted for 8%, and those aged 51 years and above accounted for 0.9%. In terms of education, those with a high school education or below accounted for 23.9% of the total sample, those with a vocational degree accounted for 70.8%, and those with a bachelor's degree or above accounted for 5.3% (see Table 1).

## Ethical statement

The studies involving human participants were reviewed and approved by the Renmin University of China. Written informed consent to participate in this study was provided by the participants.

## Measures

We used widely accepted scales with high consistency and reliability to measure major variables, such as red tape, emotional labor, and negative emotions. We translated the scales from English to Chinese and adapted the context to the participants' work settings.

Red tape was measured using the Red Tape Scale developed by Borst (2019) [8], which consists of five items, such as "Some rules and procedures sometimes make it difficult for me to complete my work well."Although our study focuses on COVID – related procedures, this general – purpose measure was chosen because it can comprehensively capture various aspects of red tape in the workplace. At the time of data collection, some of the COVID – related restrictions were still in effect, but in a more relaxed form. For example, customer entry registration was still required in some stores, but the frequency and complexity had decreased. These remaining restrictions still had an impact on employees' work, and the Red Tape Scale was able to reflect the overall perception of red tape, including both COVID – related and other work-place – specific rules that employees considered burdensome Cronbach's alpha was 0.883. The survey instruments were first translated from English to Chinese using a forward-backward translation method. Two bilingual experts conducted the initial translation and a separate group of experts reviewed the back-translation for semantic consistency. A pilot study with 35 frontline retail employees was conducted to assess item clarity and contextual relevance, leading to minor adjustments in wording.

**Table 1.**  *Means, standard deviations, and correlations of variables.*

|  | M | SD | 1 | 2 | 3 | 4 | 5 | 6 | 7 | 8 | 9 |
|---|---|---|---|---|---|---|---|---|---|---|---|
| **1. Gender** | 0.56 | 0.50 | — |  |  |  |  |  |  |  |  |
| **2. Age** | 1.83 | 1.01 | 0.14** | — |  |  |  |  |  |  |  |
| **3. Education** | 1.28 | 0.72 | −0.01 | −0.27*** | — |  |  |  |  |  |  |
| **4. Tenure** | 2.35 | 1.55 | 0.15** | 0.89*** | −0.29*** | — |  |  |  |  |  |
| **5. Red tape** | 3.59 | 0.79 | 0.09 | 0.12* | −0.14** | 0.10 | (0.883) |  |  |  |  |
| **6. Negative emotions** | 3.92 | 0.71 | 0.12* | 0.08 | −0.06 | 0.08 | 0.48*** | (0.880) |  |  |  |
| **7. Service orientation** | 3.49 | 0.69 | 0.12* | 0.08 | −0.01 | 0.06 | 0.26*** | 0.35*** | (0.827) |  |  |
| **8. Surface acting** | 3.02 | 0.93 | −0.06 | 0.01 | −0.12* | −0.02 | −0.18*** | −0.20*** | −0.02 | (0.882) |  |
| **9. Deep acting** | 2.40 | 0.91 | −0.04 | −0.14** | 0.12* | −0.16** | −0.35*** | −0.36*** | −0.10* | 0.43*** | (0.867) |

Negative emotions were measured using the Negative Emotions Scale developed by Watson (1988) [38], which consists of four items, such as "When encountering impolite behavior from the public, I feel anxious." Although this scale may appear to measure trait negative affect to some extent, in the context of our study, we believe it is still appropriate. In a retail work environment during the post – pandemic period, employees' negative emotions at work are often influenced by long – term factors such as the overall work pressure and the perception of red tape. These factors can make employees' negative emotions at work relatively stable, similar to trait – like characteristics. Moreover, we measured negative emotions in relation to work – related situations, such as employees' reactions to customer behavior and work procedures. Future research could consider using more state – specific negative emotion measures to cross – validate our results. Cronbach's alpha was 0.880.

Surface acting and deep acting scale were measured each with three items developed by Brotheridge and Lee(2002) [16]. A sample item of surface acting is " Pretend to have emotions that I don't really have". An example item of deep acting is "Really try to feel the emotions I have to show as part of my job". Cronbach's alphas of surface acting and deep acting were 0.882 and 0.867, respectively.

Customer orientation was measured using the six-item scale adopted by Mullins and Syam(2014) [39], developed by Thomas et al. (2001) [40]. After consulting relevant experts and considering the characteristics of public organizations, modifications were made to measure subordinates' customer orientation. A sample item is "I try to figure out what the public's needs are."In this study, Cronbach's alpha coefficients were 0.939 and 0.891, respectively.

All items were rated on a five-point Likert scale, with "1" to "5" representing five levels from "strongly disagree" to "strongly agree."

## Results

Before testing the hypotheses, confirmatory factor analysis (CFA) was conducted, which showed that a five-factor model fit the data acceptably: $\chi^2/df = 2.63$, $p < 0.001$, CFI = 0.942, TLI = 0.931, SRMR = 0.0516, and RMSEA = 0.0641 (see Table 2).

In our data analysis, we calculated confidence intervals using the [describe the specific method used, e.g., bootstrap method]. The confidence intervals provide a range within which the true population parameter is likely to lie. For model fit criteria, we used multiple indicators. In the confirmatory factor analysis, we considered the $\chi^2/df$ ratio, Comparative Fit Index (CFI), Tucker - Lewis Index (TLI), Standardized Root Mean Square Residual (SRMR), and Root Mean Square Error of Approximation (RMSEA). A $\chi^2/df$ ratio between 1–3, CFI and TLI values greater than 0.9, SRMR less than 0.08, and RMSEA less than 0.08 are generally considered acceptable fit indicators. We analyzed surface acting and deep acting separately rather than within a unified path analysis because surface acting and deep acting represent different dimensions of emotional labor with distinct underlying mechanisms. Surface acting involves external emotional regulation, while deep acting focuses on internal emotional alignment. Analyzing them separately allows us to more precisely explore how red tape, negative emotions, and customer orientation affect each dimension, providing a more in – depth understanding of the complex relationship between these variables.

**Table 2. Results of confirmatory factor analysis.**

|  | $\chi^2$ | df | CFI | TLI | SRMR | RMSEA |
|---|---|---|---|---|---|---|
| RE, NE, DA, SA, CO | 421 | 160 | 0.942 | 0.931 | 0.0516 | 0.0641 |
| RE + NE, DA, SA, CO | 1080 | 164 | 0.796 | 0.764 | 0.0817 | 0.119 |
| RE + NE + DA, SA, CO | 2067 | 206 | 0.634 | 0.59 | 0.123 | 0.151 |
| RE + NE + CO, DA + CA | 2577 | 208 | 0.534 | 0.483 | 0.133 | 0.17 |
| RE + NE + DA + SA + CO | 3254 | 209 | 0.401 | 0.338 | 0.157 | 0.192 |

*Notes*: RE = red tape, NE = negative emotions, DA = deep acting, SA = surface acting, CO = customer orientation.

We first used multiple regression to test the relationships among red tape, negative emotions, and each type of acting behavior (see Table 3). After adjusting for demographic variables, namely gender, age, education and tenure, we found that red tape was significantly and positively related to negative emotions (β = 0.47, p < 0.01), and red tape was significantly negatively related to surface acting (β = −0.20, p < 0.01) and deep acting (β = −0.34, p < 0.01). Hypotheses 1 and 2 were therefore supported. After controlling negative emotions, the relationship between red tape and surface acting was less significant (β = −0.14, p < 0.05), indicating a partial mediating effect of negative emotions between red tape and surface acting. Thus, Hypothesis 3 was supported. We also found that the coefficient of red tape on deep acting decreased with negative emotions included in the model, signifying that negative emotions partially mediate the path from red tape to deep acting. Thus, Hypothesis 4 was supported.

In addition, service orientation negatively moderated the relationship between negative emotions and surface acting (β = −0.07, p < 0.05) and deep acting (β = −0.11, p < 0.001), such that higher levels of service orientation strengthened the adverse impact of negative emotions on the surface and deep acting. We further plotted the moderating effects of service orientation by showing the effects of negative emotions on surface acting (Fig 2) and deep acting (Fig 3) at different levels of service orientation. For individuals with higher customer orientation, the negative emotions caused by red tape had less impact on surface acting but did affect deep acting. This means that the higher the customer orientation and the more a person dislikes red tape, the more it will reduce their deep acting. Thus, Hypotheses 5 and 6 were supported.

We further tested the moderated mediation models (see Table 4). The results showed that the mediation effects of negative emotions between red tape and both acing behaviors were strengthened as the levels of service orientation increased. Specifically, when employees were in high service orientation, the indirect effect size of red tape on surface acting via negative emotions was significantly negative (B = −0.13, SE = 0.05, 95% CI = [−0.2432, −0.0490]), When employees were low in service orientation, the indirect effect size of red tape was not significant (B = −0.05, SE = 0.04, 95% CI = [−0.1412, 0.0252]). The indirect effect of red tape on deep acting became lower as service orientation increased from low, medium to high levels (B = −0.08, −0.14, −0.20, respectively). However, the moderated mediation effect of negative emotions between red tape and surface acting was not significant (B = −0.06, SE = 0.04, 95% CI = [−0.1289, 0.0173]), and that between red tape and deep acting was significant (B = −0.09, SE = 0.03, 95% CI =

**Table 3. Multiple regression results.**

| | Negative emotions | | Surface acting | | | | Deep acting | | | |
|---|---|---|---|---|---|---|---|---|---|---|
| | Model 1 | Model 2 | Model 3 | Model 4 | Model 5 | Model 6 | Model 7 | Model 8 | Model 9 | Model 10 |
| Intercept | | | | | | | | | | |
| Gender | 0.09 | 0.05 | −0.03 | −0.02 | −0.01 | −0.02 | 0.00 | 0.03 | 0.04 | 0.04 |
| Age | 0.03 | −0.05 | 0.16 | 0.19 | 0.18 | 0.18 | 0.00 | 0.06 | 0.05 | 0.05 |
| Education | −0.05 | 0.00 | −0.12* | −0.14* | −0.14* | −0.15** | 0.09 | 0.05 | 0.05 | 0.04 |
| Tenure | −0.05 | −0.01 | −0.11 | −0.12 | −0.12 | −0.12 | −0.06 | −0.09 | −0.09 | −0.09 |
| Red tape | | 0.47*** | | −0.20*** | −0.14* | −0.15* | | −0.34*** | −0.23*** | −0.23*** |
| Negative emotions | | | | | −0.12* | −0.16 | | | −0.22*** | −0.26 |
| Customer orientation | | | | | | 0.09* | | | | 0.05*** |
| Negative emotions × Service orientation | | | | | | −0.07* | | | | −0.11*** |
| **R²** | 0.04 | 0.25 | 0.05 | 0.08 | 0.09 | 0.11 | 0.05 | 0.16 | 0.20 | 0.22 |
| Adjusted R² | 0.02 | 0.24 | 0.03 | 0.07 | 0.08 | 0.09 | 0.04 | 0.14 | 0.18 | 0.20 |
| ΔR² | | 0.22 | | 0.04 | 0.01 | 0.01 | | 0.11 | 0.04 | 0.03 |
| **F** | 2.90* | 21.79*** | 3.77** | 5.90*** | 5.76*** | 5.36*** | 3.95*** | 12.15*** | 13.44*** | 12.24*** |

*Notes* Standardized regression coefficients are reported. *p < 0.05; **p < 0.01; ***p < 0.001.

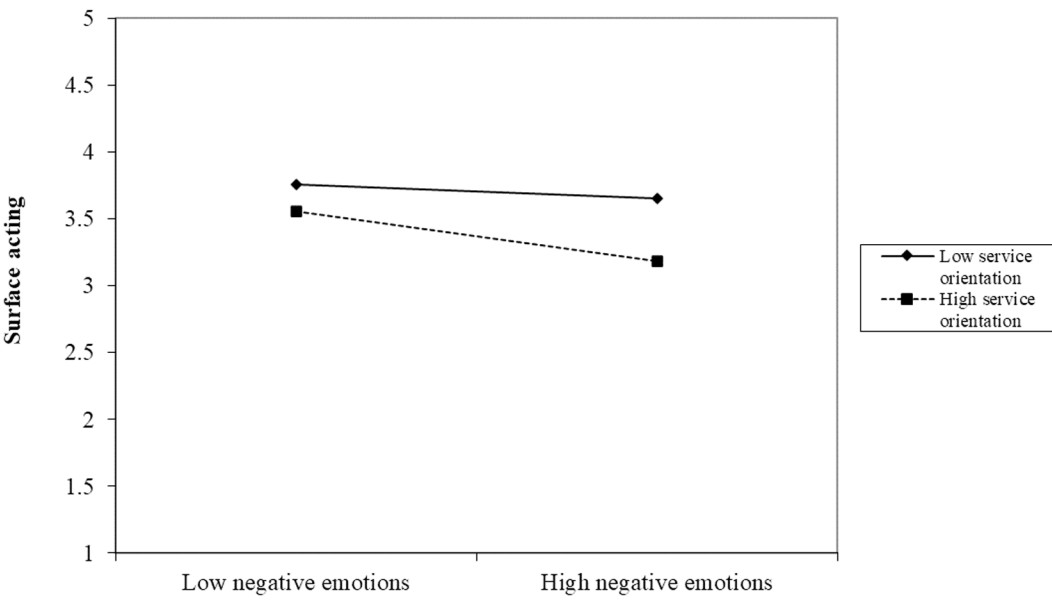

**Fig 2. The moderating role of customer orientation between negative emotions and surface acting.**

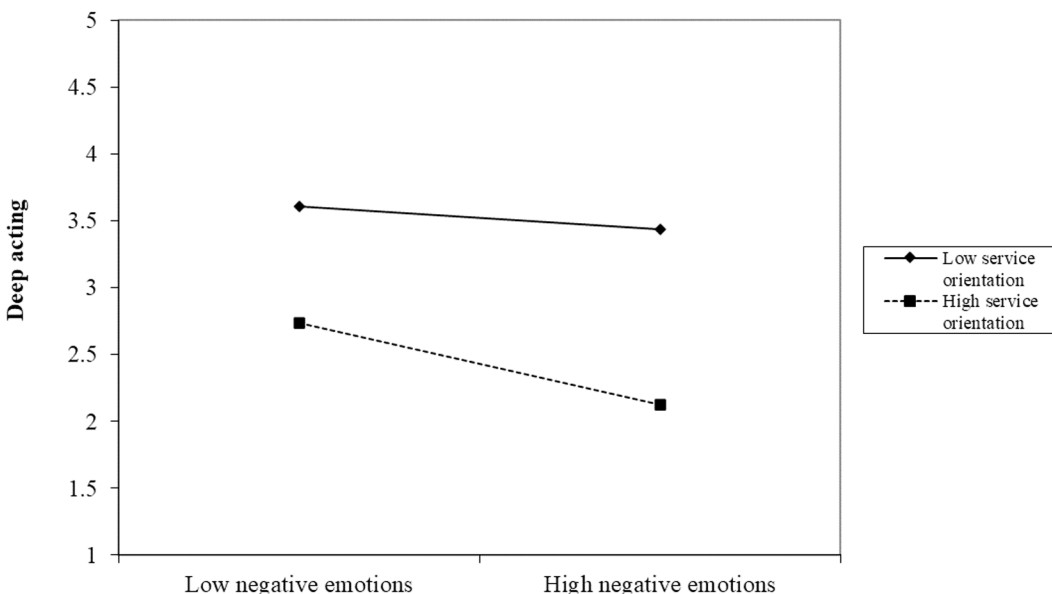

**Fig 3. The moderating role of customer orientation between negative emotions and deep acting.**

[−0.1494, −0.0263]. Thus, we concluded that the moderated mediation model was supported only for the path from red tape to deep acting.

It should be noted that the confidence interval for the conditional indirect effect of red tape on surface acting via negative emotions at high customer orientation includes zero. This suggests that the moderated mediation effect is not statistically robust in this condition, and readers should interpret this result with caution. The lack of significance may be attributed to sample size limitations or contextual variability.

**Table 4. Conditional indirect effects of negative emotions on surface/deep acting at different values of the service orientation.**

| Outcomes | Moderator: Customer orientation | Conditional indirect effect | | | Moderated mediation effect | | |
|---|---|---|---|---|---|---|---|
| | | B | SE | 95% | B | SE | 95% |
| Surface acting | Low | −0.22 | 0.09 | [-0.40, -0.04] | 2.80 | −0.04 | [-0.14, 0.04] |
| | Median | −0.18 | 0.07 | [-0.31, -0.05] | 3.49 | −0.09 | [-0.18, -0.02] |
| | High | −0.14 | 0.08 | [-0.30, 0.02] | 4.17 | −0.13 | [-0.25, -.05] |
| Deep acting | Low | −0.33 | 0.08 | [-0.50, -0.17] | −0.07 | 0.04 | [-.15, -0.00] |
| | Median | −0.28 | 0.06 | [-0.40, -0.16] | −0.14 | 0.04 | [-0.22, -0.08] |
| | High | −0.22 | 0.08 | [-0.36, -0.07] | −0.21 | 0.05 | [-0.32, -0.13] |

Table 3 presents the multiple regression results. Model 1 is the baseline model for predicting negative emotions, including only demographic variables. Model 2 adds the independent variable'red tape' to predict negative emotions. Model 3 is the baseline model for predicting surface acting, with only demographic variables. Model 4 adds'red tape' to predict surface acting. Model 5 further adds 'negative emotions' to explore its mediating effect on the relationship between red tape and surface acting. Model 6 includes 'customer orientation' and the interaction term 'negative emotions × service orientation' to test the moderating effect. The models for deep acting (Model 7 – Model 10) follow a similar logic. Each variable in the table is clearly labeled, with standardized regression coefficients reported to show the strength and direction of the relationships

## Discussion and implications

### Summary of the main findings

Based on COR theory and the JD-R model, this study consisted of an in-depth investigation into the relationship between organizational red tape and emotional labor in enterprises, as well as the roles of negative emotions and customer orientation in this relationship. The empirical analysis found that organizational red tape negatively affected the surface and deep acting of enterprise employees. In addition, negative emotions mediated the relationship between red tape and surface acting; red tape suppressed the surface and deep acting of enterprise employees by increasing negative emotions. Third, customer orientation negatively regulated deep acting but has no moderating effect on surface acting. Employees with high customer orientation may adopt alternative regulation strategies, such as cognitive reappraisal, instead of surface acting.

### Theoretical implications

Firstly, this study shed light on the impact of red tape on employees' emotional labor within Chinese enterprises. In the past, research regarding red tape typically centered on internal government organizations, with scant attention given to enterprises. Nevertheless, numerous enterprises in the service industry also encounter the red – tape phenomenon. This has an impact on employees' emotional labor, and subsequently affects their service attitude and the results of the services they provide. The conclusions of this study offer a reference that can assist Chinese enterprises in dealing with red tape more effectively, ultimately contributing to an improvement in enterprise efficiency. Employees with high customer orientation may adopt alternative regulation strategies, such as cognitive reappraisal, instead of surface acting.

Second, this study focused on the issue of red tape using quantitative research, thereby demonstrating the hidden drawbacks of red tape within enterprises and challenging overly optimistic perceptions of enterprises. Enterprises have always been guided by efficiency and profit, but many enterprises have considerable red tape, which affects employees' emotions and work behavior. These conclusions compensate for the lack of enterprise-level research in the Chinese context and illustrate the importance of the red tape issue in enterprise organizations.

Thirdly, this research broadens the application of the Conservation of Resources (COR) theory and the behavioral control theory in the management and psychology domains. These two theories postulate that when individuals sense a threat, they take proactive steps to regulate their behavior to prevent potential losses. Our study reveals that in the context of Chinese enterprises, negative emotions among employees and challenges related to emotional labor are significant concerns. Precisely, as the extent of red tape in an enterprise becomes more severe, employees tend to experience more intense negative emotions. These negative emotions then have a detrimental impact on their emotional labor. By demonstrating this relationship, our research offers theoretical underpinnings that can support the growth and efficiency enhancement of Chinese enterprises.

Fourth, we have expanded the role of customer service orientation in the relationship between organizational red tape and proactive employee behavior. Previous research predominantly posits that customer service orientation can effectively alleviate the negative emotions induced by red tape, enhancing employees' surface acting and deep acting. However, our findings suggest that employees with a higher degree of customer service orientation experience increased negative emotions due to red tape. This is because they are often more inclined to provide better service to customers rather than adhering to bureaucratic procedures, which subsequently reduces efficiency.Employees with high customer orientation may adopt alternative regulation strategies, such as cognitive reappraisal, instead of surface acting.

## Practical implications

This study has significant implications for the management practices of enterprises in China. First, enterprises should reflect on the disadvantages of their procedures and develop corresponding solutions. For example, companies could curb any increase in red tape. The danger of excess red tape should not be underestimated during the development of Chinese enterprises. Enterprises should focus on reducing unnecessary internal red tape, adopting more efficient and scientific methods, and avoiding obstacles. Second, government departments should actively participate in reducing red tape. To cooperate with government red tape, some enterprises increased their burden, which is not consistent with the concept of simplifying administration and delegating power and even more inconsistent with the ultimate goal of enterprises of improving efficiency and profitability. Third, it is essential to pay attention to the emotions and emotional labor of employees in enterprises. Within the collective, enterprises should pay more attention to employees' emotions, facilitating them experiencing as few negative emotions as possible and thus working more efficiently. Therefore, enterprises should continuously improve their internal environment, enhance their image, and thereby win the trust and favor of their employees.

## Limitations and future directions

First, although this study employed experimental research and collected survey data at multiple time points, using multiple sources to ensure the robustness of the conclusions, there may have been some limitations regarding sample selection. Our study collected data from frontline retail employees in Beijing. Although this sample provides valuable insights into the impact of red tape on emotional labor in a specific context, the generalizability of our findings may be limited. Beijing has its own unique economic, cultural, and social characteristics, and the retail industry in Beijing may also have different operational models compared to other regions. However, the issues of red tape and emotional labor are common in the service industry. Future research could expand the sampling scope to include employees from different cities and regions, as well as different types of service industries, to further verify the applicability of our findings. The contextual specificity of post-pandemic Beijing retailing may also limit the generalizability of the moderated mediation results. Future studies should expand the sample to cover diverse cities and sectors, and incorporate multi-source data (e.g., managerial ratings or customer satisfaction records) to further validate the observed relationships. Additionally, we can conduct cross – cultural studies to explore whether cultural differences affect the relationship between red tape, emotional labor, negative emotions, and customer orientation.The data were exclusively gathered in Beijing, making it uncertain whether the

findings of our study can be extrapolated to other urban centers or more extensive regions. Subsequent research should consider broadening the scope of sampling to include additional geographical locations, which could further verify the applicability of the conclusions to other types of organizations. Second, this study only considered the impact of red tape at the individual level without simultaneously considering the synergistic effects at the individual and organizational levels. Future studies could use cross-level research methods to further reveal the impact of different organizational red tape on individuals, thereby enriching our understanding of the research model. Finally, the data were self-reported and obtained from questionnaires administered to frontline retail staff at shopping malls. This raises the issue of common method bias. Future studies could use leaders' evaluations of their employees' proactive behavior and customer satisfaction as data sources to address this issue.

## Supporting information

**S1 File. Supporting Information (S1_File.xlsx) to ensure transparency and replicability.**
(XLSX)

## Author contributions

**Conceptualization:** Qin Qiang, Kaixin Wang.

**Data curation:** Kaixin Wang.

**Formal analysis:** Kaixin Wang.

**Methodology:** Jianxin Lai.

**Writing – original draft:** Jianxin Lai.

**Writing – review & editing:** Qin Qiang.

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
