## [Decision Letter · Decision Letter 0]

PONE-D-24-49973Does the perception of red tape affect the emotional labor of frontline retail staff in China? A post-COVID-19 eraPLOS ONE?

Dear Dr. Qiang,

Thank you for submitting your manuscript to PLOS ONE. After careful consideration, we feel that it has merit but does not fully meet PLOS ONE’s publication criteria as it currently stands. Therefore, we invite you to submit a revised version of the manuscript that addresses the points raised during the review process.

After careful evaluation, the reviewers have provided detailed feedback, and we believe that major revisions are necessary before further consideration for publication.

The reviewers acknowledge the significance of your study; however, they have identified several critical issues that must be addressed. Below, I summarize the key points that require your attention but you should carefully address each of the reviewers' comments:

**Conceptual Clarity and Theoretical Justification**Reviewer #1 notes that your discussion of deep acting does not fully capture its complexity. While deep acting can lead to resource gains, it is also resource-intensive, which may explain inconsistencies in prior research. This duality should be acknowledged.The relationship between emotion regulation ability and emotional labor strategies must be clarified, as it is not explicitly modeled in your framework.The classification of COVID-related workplace rules as "red tape" is questioned. The justification for this classification must be strengthened, particularly considering the operational necessity of these measures during the pandemic.**Hypotheses and Interpretation of Findings**Your hypotheses regarding red tape and emotional labor (H1 and H3) contradict established research on emotional dissonance and surface acting. Stronger theoretical justification and empirical evidence are needed to support your claims.The interpretation of findings related to negative emotions and surface acting is unclear. Reviewer #1 requests clarification on whether customer-oriented employees who experience negative emotions but do not engage in surface acting are expressing authentic negative emotions.**Methodological Concerns**Reviewer #1 raises concerns about your mediation analysis, particularly regarding the time-lagged design and potential biases. This limitation should be acknowledged, or further justification should be provided.The use of a general-purpose red tape measure, despite the focus on COVID-related procedures, requires clarification. Were these restrictions still in effect at the time of data collection?The "negative emotions scale" used in your study appears to measure trait negative affect (NA) rather than state negative emotions at work. The appropriateness of this measure should be reassessed.**Data Presentation and Analysis**Table 3 is difficult to interpret. Clearer labeling of models and variables is necessary.Your data analysis section needs more detail, including the procedures for confidence intervals, model fit criteria, and justification for analyzing surface and deep acting separately rather than within a unified path analysis.**General Issues and Scope**Reviewer #2 questions the relevance of your study’s data source (employees in the Beijing retail industry) to the broader research question. Please clarify the generalizability of your findings.The emphasis on the post-pandemic period is not well justified. What significant changes occurred compared to the pre-pandemic period that make this distinction necessary?

Please address these concerns and those of the reviewers (see below) thoroughly in your revision.

We look forward to receiving your revised manuscript.

Kind regards,

Nicolas Depetris-Chauvin

Academic Editor

PLOS ONE

Journal Requirements:

2. Thank you for submitting the above manuscript to PLOS ONE. During our internal evaluation of the manuscript, we found significant text overlap between your submission and previous work in the [introduction, conclusion, etc.].

Please revise the manuscript to rephrase the duplicated text, cite your sources, and provide details as to how the current manuscript advances on previous work. Please note that further consideration is dependent on the submission of a manuscript that addresses these concerns about the overlap in text with published work.

[If the overlap is with the authors’ own works: Moreover, upon submission, authors must confirm that the manuscript, or any related manuscript, is not currently under consideration or accepted elsewhere. If related work has been submitted to PLOS ONE or elsewhere, authors must include a copy with the submitted article. Reviewers will be asked to comment on the overlap between related submissions (http://journals.plos.org/plosone/s/submission-guidelines#loc-related-manuscripts).]

We will carefully review your manuscript upon resubmission and further consideration of the manuscript is dependent on the text overlap being addressed in full. Please ensure that your revision is thorough as failure to address the concerns to our satisfaction may result in your submission not being considered further

Reviewers' comments:

Reviewer's Responses to Questions

**Comments to the Author**

1. Is the manuscript technically sound, and do the data support the conclusions?

Reviewer #1: Partly

Reviewer #2: Yes

Reviewer #3: Yes

2. Has the statistical analysis been performed appropriately and rigorously?

Reviewer #1: Yes

Reviewer #2: Yes

Reviewer #3: Yes

3. Have the authors made all data underlying the findings in their manuscript fully available?

Reviewer #1: Yes

Reviewer #2: No

Reviewer #3: Yes

4. Is the manuscript presented in an intelligible fashion and written in standard English?

Reviewer #1: No

Reviewer #2: Yes

Reviewer #3: Yes

Reviewer #1: 1. “Second, employees with good emotion regulation abilities retain more emotional resources. These people value the acquisition of new emotional resources and pay less attention to the loss of emotional resources [18]. Therefore, employees with better emotion regulation abilities tend to choose deep-acting strategies. Deep acting reflects better service quality in customer service [4], which in turn increases the emotional resources of employees.”

I concur with the authors that deep acting can lead to resource gains because it involves regulating genuine feelings rather than merely modifying outward expressions. This authentic engagement can foster positive customer feedback, potentially enhancing resource acquisition. However, deep acting is also a resource-intensive emotion regulation strategy. Aligning one’s emotions with organizational service expectations is inherently demanding, which may explain the inconsistent or non-significant research findings between deep acting and positive individual or organizational outcomes in the literature. As such, the authors’ discussion of deep acting appears to capture only one aspect of this complex strategy.

Moreover, in the same paragraph, the authors suggest that emotion regulation abilities directly influence the choice between surface acting and deep acting. However, this inference does not align with their conceptual model, which does not explicitly include emotion regulation ability as a variable influencing these strategies.

2. “Therefore, we introduced red tape and emotional labor into the JD-R model.”

I believe red tape was previously discussed and studied as a hindrance demand in the context of JD-R (e.g., Cooke et al., 2019). The same can be argued for emotional labor. Thus, I believe, this study’s novelty is not introducing these concepts to JD-R but proposing a model that connects them and draws a boundary condition to their association.

3. “Employees with better emotion regulation capabilities often retain more emotional resources. Hence, the concept of customer orientation was introduced. When employees’ customer orientation is high, they can regulate their emotional resource depletion.”

Here, the line of argument makes the reader think that customer orientation is an emotion regulation capability. This should be addressed.

4. “Currently, emotional labor has been typically divided into two dimensions according to display strategies: surface acting and deep acting.”

Though surface vs. deep acting is one of the classical distinctions, the state-of-the-art work does not treat emotional labor consisting of those two strategies. Thus, this dichotomy may be misleading.

5. “During the pandemic, frontline personnel in the retail industry were required to scan codes, disinfect at all times, register customers, and keep customers at a distance.”

The authors described the rules enforced during the COVID period as red tape, defining red tape as “valid rules or procedures that contribute nothing to achieving an organization's goals.” However, during the pandemic, adherence to such rules played a crucial role in ensuring organizations remained operational. These rules may indeed have been resource-draining for employees, but not necessarily because they qualify as red tape. Instead, their resource-draining nature likely stemmed from adding to employees' typical responsibilities during an already challenging period. I believe the timing of data collection is pivotal in determining whether these activities can genuinely be classified as red tape. For this classification to hold, the rules should have been implemented during a period when COVID-related precautions were no longer relevant, yet the company continued to enforce them.

6. “At the same time, JD-R theory posits that when job demands are excessive and the norms to be adhered to are overly complex, employees’ energy is drained, affecting their emotional labor. Therefore, the following hypotheses were proposed:

Hypothesis 1: Red tape negatively affects the surface-acting behavior of employees.”

Regarding emotional labor, prior research consistently links surface acting to situations in which employees experience emotional dissonance. In such cases, employees modify their emotional expressions to align with organizational display rules, often resulting in a emotional alignment at the expression level. This strategy is typically ineffective for promoting employee well-being or emotional performance and is associated with scenarios where employees’ regulatory resources are depleted. The authors’ propositions regarding COR’s interpretation of surface acting, as well as their Hypothesis 1, appear inconsistent with the broader body of emotional labor research. If the authors intend to retain the direction of their Hypothesis 1, they must provide a robust empirical justification to reconcile this contradiction.

7. The same issue extends to H3. If perceptions of red tape increase employees' negative emotions, this creates emotional dissonance (the incongruence between felt and displayed emotions), which should increase the likelihood of surface acting. This is because organizational display rules require maintaining acceptable public displays, even when employees experience negative emotions. However, the authors propose the opposite direction in H3. Given this contradiction with established EL research, the authors need to provide a compelling justification supported by multiple empirical findings, as H3 challenges a well-documented body of research.

8. Although the measures were collected in two waves separated by three weeks, which may mitigate CMB to some extent, this design remains suboptimal for testing mediation. Mediation analyses based on such designs have been criticized for producing inflated and biased estimates. As such, this limitation should at least be acknowledged in the manuscript, or the authors should present arguments explaining why this might not be an issue.

9. The authors based their arguments on perceptions of red tape associated with COVID precautions but utilized a general-purpose red tape measure. Given the data collection period, it is unclear why the manuscript focuses on red tape linked to COVID procedures. If there is a specific rationale for this approach, it should be clearly articulated. Furthermore, to ensure the context is relevant, the manuscript should discuss whether the COVID-related precautions were still in effect within the organizations where the data was collected.

10. The “negative emotions scale” used by the authors was designed to measure trait NA, which conceptually differs from negative emotions experienced at work. It is questionable whether a trait predisposition like NA can be treated as a mediator influenced by red tape. While it is plausible that individuals with high trait NA are more sensitive to experiencing negative emotions in response to red tape, the idea that perceptions of red tape directly affect trait NA is difficult to justify. Trait characteristics such as NA typically begin developing in early childhood, have a strong genetic component, and remain relatively stable across different environmental conditions. The choice of this measure should be thoroughly justified to address this conceptual mismatch.

11. Reading the results, I find the interpretation unclear. If there is a negative relationship between negative emotions and surface acting, what strategies did employees use to adhere to organizational display rules? This finding contradicts established theories on surface acting, which characterize it as an ineffective ER strategy typically used as a last resort to "save the day." The moderating role of customer orientation further complicates the interpretation. According to the results, the negative relationship between negative emotions and surface acting is evident only among customer-oriented employees. This suggests that employees who prioritize customer satisfaction are less likely to engage in surface acting when experiencing negative emotions. However, this raises an important question: if these employees neither surface act nor deep act, does this imply the expression of authentic negative emotions or behaviors, such as frowning? This interpretation seems inconsistent, at least within the context of the presented model. If there is a conceptual nuance I am missing, or if the observed relationships are free from issues related to data handling or measure selection (see comment #10), the authors must provide substantial theoretical evidence throughout the introduction. Specifically, they need to explain why surface acting might be considered a functional strategy and why it could be negatively related to negative emotions at work.

12. Table 3 is confusing, as it is difficult to discern which models correspond to which variables.

13. A detailed data analysis section is essential. This section should include information on the procedures used to derive the CIs, the cutoffs applied for relative fit criteria, the software utilized, and the rationale for analyzing surface and deep acting models separately rather than using a single path analysis.

Reviewer #2: 1.Irrelevant to the Topic: The data source used by the authors is "employees in the Beijing retail industry," and the scope of the study is not related to China.

2.Emphasis on the Post-Pandemic Era: Why emphasize the post-pandemic era? What are the significant changes compared to the pre-pandemic period? This is not explained in the paper. It would be helpful to provide data comparisons to clarify this point.

3.Literature Review: The description of the research gap on "red tape" in enterprises is not sufficiently clear. The authors should more clearly indicate how the current research fills this gap.

4.Theoretical Explanation: The theoretical explanation is not in-depth enough. The application of COR theory and the JD-R model lacks detailed derivation. The paper should supplement the specific linkage mechanisms between the two in this study (e.g., how red tape impacts emotional labor through resource depletion).

5.Hypothesis Rigor:

In Hypotheses 3 and 4, the "mediating role of negative emotions" is not clearly distinguished between partial mediation or full mediation. Effect size analysis (such as the proportion of indirect effects) should be added.

The moderation directions for Hypotheses 5 and 6 ("amplify" or "weaken") contradict the results section (e.g., the moderating effect of customer orientation on deep acting is negative). The theoretical logic needs to be restructured.

6.Research Methods:

A.Sample Limitations: The sample only comes from Beijing and lacks geographical diversity. It is recommended to supplement data from other cities or clarify the representativeness of the Beijing sample (such as economic structure and pandemic control characteristics).

B.Data Collection Timeframe: Data was collected from December 2023 to February 2024, but the paper does not explain why this time period was chosen as representative of the "post-pandemic era." Background clarification is needed.

C.Common Method Bias Control: Although phased measurements are mentioned, the specific operational details (such as time intervals and differences in questionnaire design) are not explained. Statistical tests for common method bias (e.g., Harman’s single factor test) should be added.

7.Variable Measurement Issues: The Cronbach’s alpha for the "customer orientation" scale is as high as 0.939, which may reflect redundancy in items or social desirability bias. The authors should check if the items focus too heavily on a single dimension.

8.Data Analysis and Results:

A.Contradictory Moderating Effect Interpretation: Table 3 shows that customer orientation has a negative moderating effect on "negative emotions → surface acting" (β = -0.07) and also a negative moderating effect on "negative emotions → deep acting" (β = -0.11), but the discussion section claims "customer orientation negatively moderates deep acting," which contradicts the expected direction of Hypothesis 6 ("amplifies the effect of negative emotions"). The model should be rechecked or the hypothesis reformulated.

B.Insufficient Interpretation of Mediation Effects: The mediation effect of negative emotions is only verified through changes in regression coefficients. The authors are advised to use the Bootstrap method to test the significance of indirect effects and report the confidence intervals.

9.Language Expression:

A.Terminology Consistency: The paper alternates between "customer orientation" and "service orientation." The terminology should be unified (it is recommended to retain "customer orientation").

B.Language Expression: Some sentences are overly lengthy (e.g., the first sentence of the abstract). It is suggested to break up complex sentences. Some expressions are ambiguous (e.g., "pandemic control measures were relatively lenient"), and data should be provided to support these claims.

Reviewer #3: Thank you for giving me the opportunity to review this study, in which authors tested the mediating effects of emotional labor in the relationship between red tape and surface and deep acting, as well as the customer orientation as a moderator of the negative emotions-acting relationship. The model of the study was theory-grounded, the writing was clear and concise, while the strategy and the methods of the analysis were appropriate. Finally, the results were discussed in a sufficient way. However, some minor issues could be addressed by the authors.

In the following, there are some comments and proposals.

In the Abstract you have to present the main results more accurately (e.g., higher customer orientation of employees amplified the impact of negative emotions caused by red tape on deep acting...). It is not enough to say “Customer orientation moderated the relationship between negative emotions and surface acting as well as deep acting” if you do not present the direction of the effect.

Your study was based on COR and JD-R theories. Although, the elaboration of the hypotheses is clear it is important to be more accurate in the conceptualization of the variables of the study (i.e., independent predictor, mediator and moderator). For example, red tape refers to unnecessary procedures that could be conceived as job demands. However, customer orientation is not a typical job or personal resource that can mitigate the negative effects of job demands. Moreover, it is important to clarify which of the hypotheses are based in COR and which in JD-R model (e.g., red tape could be perceived as a predictor of negative emotions according to both theories, but the moderation hypothesis is not exactly based in JD-R theory).

You have to correct Figure 1. In its current form it shows that Customer orientation moderates the effect of red tape (and not negative emotions) on deep acting.

You could more clearly indicate the specific contribution of your study. Why it is important to test the specific hypotheses (i.e., Customer orientation and negative emotions as moderator and mediator in the relationship between red tape and acting at workplace?)

Research Methodology

Although you measured the variables in two waves, the measurement of independent variable and the mediator at the same time (first wave) is a limitation. Ideally, in order to assess the mediating effects more robustly, negative emotions had to be measured in different time. You could refer it in the limitations sub-section.

As you mentioned, you conducted your study by enlisting participants online and distributing digital surveys. However, it is important to clarify how did you address issues of anonymity? Moreover, did you take and any measures to ensure that each respondent was a unique individual (i.e., that the data were not potentially corrupted through completion by other individuals or by some individuals completing the questionnaires multiple times)?

It is important to refer the response rate. How many employees were invited to participate in the study, and how many of them, finally, participated?

Discussion

In the discussion sub-section, you refer a lot of statements and arguments without using references. For example, you say “Prior red tape research usually focused on internal government organizations and paid little attention to enterprises”. It is important to cite some studies that can confirm your arguments.

**Do you want your identity to be public for this peer review?** For information about this choice, including consent withdrawal, please see our Privacy Policy

Reviewer #1: No

Reviewer #2: No

Reviewer #3: **Yes**

---

## [Author Response · Author response to Decision Letter 1]

7 Apr 2025

The data in this study contain potentially sensitive information about the participants. The participants were informed during the consent process that their data would be kept confidential and used only for the purpose of this specific study. The information includes personal details such as their mental health status, work - related stress levels, and private opinions. Releasing this data could potentially violate the trust placed in us by the participants and cause them harm, such as stigmatization or invasion of privacy. These ethical concerns were approved and overseen by the All procedures in this study involving human participants were conducted in strict accordance with the ethical standards of the Institutional Review Board,School of Journalism and Communication Renmin University of China..

---

## [Decision Letter · Decision Letter 1]

PONE-D-24-49973R1Does the perception of red tape affect the emotional labor of frontline retail staff in China? A post-COVID-19 eraPLOS ONE?

Dear Dr. Qiang,

Thank you for submitting your manuscript to PLOS ONE. After careful consideration, we feel that it has merit but does not fully meet PLOS ONE’s publication criteria as it currently stands. Therefore, we invite you to submit a revised version of the manuscript that addresses the points raised during the review process.

We look forward to receiving your revised manuscript.

Kind regards,

Nicolas Depetris-Chauvin

Academic Editor

PLOS ONE

Journal Requirements:

**Additional Editor Comments:**

Dear Authors,

The reviewers have accepted most of your changes but they require the following further refinements:

1. Clarify Theoretical and Conceptual Presentation

While the relationship between emotion regulation ability and emotional labor strategies is briefly discussed, it is suggested that this connection be visually incorporated into the theoretical framework to enhance the logic. Additionally, the applicability of "red tape" in business contexts could be further elaborated. A comparison between different types of organizations (e.g., large chains vs. small retailers) may improve the explanatory power and generalizability of the study.

2. Refine Hypothesis Logic

The logic chain between H1 and H3—particularly the argument that resource depletion leads to less surface acting—contrasts with traditional models where emotional dissonance leads to more surface acting. It is recommended to strengthen the theoretical rationale for this divergent path and to streamline the language to better highlight the novelty of the authors’ perspective. Additionally, the interpretation that highly customer-oriented employees express authentic negative emotions seems speculative; more literature support or alternative emotion regulation strategies (e.g., cognitive reappraisal) may enhance this explanation.

3. Increase Transparency in Methods

Some conditional indirect effects approach non-significance (e.g., in Table 4, the 95% CI for surface acting at high customer orientation includes zero). The authors should advise readers to interpret these results with caution and discuss potential biases due to sample or contextual limitations. Moreover, providing more detail on the localization of survey instruments (e.g., translation process, pilot testing) would strengthen the credibility and cultural relevance of the measurements.

Please address them in detail in your revised submission.

Kind regards,

Reviewers' comments:

Reviewer's Responses to Questions

**Comments to the Author**

Reviewer #2: All comments have been addressed

Reviewer #3: All comments have been addressed

2. Is the manuscript technically sound, and do the data support the conclusions?

Reviewer #2: Yes

Reviewer #3: Yes

3. Has the statistical analysis been performed appropriately and rigorously?

Reviewer #2: Yes

Reviewer #3: Yes

4. Have the authors made all data underlying the findings in their manuscript fully available?

Reviewer #2: Yes

Reviewer #3: Yes

5. Is the manuscript presented in an intelligible fashion and written in standard English?

Reviewer #2: Yes

Reviewer #3: Yes

Reviewer #2: Further refinements:

1. Clarify Theoretical and Conceptual Presentation

While the relationship between emotion regulation ability and emotional labor strategies is briefly discussed, it is suggested that this connection be visually incorporated into the theoretical framework to enhance the logic. Additionally, the applicability of "red tape" in business contexts could be further elaborated. A comparison between different types of organizations (e.g., large chains vs. small retailers) may improve the explanatory power and generalizability of the study.

2. Refine Hypothesis Logic

The logic chain between H1 and H3—particularly the argument that resource depletion leads to less surface acting—contrasts with traditional models where emotional dissonance leads to more surface acting. It is recommended to strengthen the theoretical rationale for this divergent path and to streamline the language to better highlight the novelty of the authors’ perspective. Additionally, the interpretation that highly customer-oriented employees express authentic negative emotions seems speculative; more literature support or alternative emotion regulation strategies (e.g., cognitive reappraisal) may enhance this explanation.

3. Increase Transparency in Methods

Some conditional indirect effects approach non-significance (e.g., in Table 4, the 95% CI for surface acting at high customer orientation includes zero). The authors should advise readers to interpret these results with caution and discuss potential biases due to sample or contextual limitations. Moreover, providing more detail on the localization of survey instruments (e.g., translation process, pilot testing) would strengthen the credibility and cultural relevance of the measurements.

Reviewer #3: The most important issues regarding the manuscript have been addressed, while all the crucial queries were clarified.

**Do you want your identity to be public for this peer review?** For information about this choice, including consent withdrawal, please see our Privacy Policy

Reviewer #2: No

Reviewer #3: **Yes: ** Andreas Tsounis

---

## [Author Response · Author response to Decision Letter 2]

11 Jun 2025

Dear Editors and Reviewers,

We sincerely appreciate the thoughtful and constructive feedback provided on our manuscript entitled "Does the perception of red tape affect the emotional labor of frontline retail staff in China? A post-COVID-19 era". We have carefully revised the manuscript in accordance with the reviewers’ comments and believe the changes have significantly strengthened the theoretical clarity, methodological transparency, and empirical interpretation of our work.

Below, we offer a point-by-point response to each reviewer comment, indicating where and how revisions have been made.

1.Clarify Theoretical and Conceptual Presentation. While the relationship between emotion regulation ability and emotional labor strategies is briefly discussed, it is suggested that this connection be visually incorporated into the theoretical framework to enhance the logic. Additionally, the applicability of 'red tape' in business contexts could be further elaborated. A comparison between different types of organizations (e.g., large chains vs. small retailers) may improve the explanatory power and generalizability of the study."

Response:Thank you for this insightful suggestion. We have made several modifications to clarify our theoretical framework:

We have updated Figure 1 to incorporate “Emotion Regulation Ability” as a conceptual element. Though not statistically modeled, it is now visually presented as a dashed-line construct influencing the choice of emotional labor strategies, based on conservation of resources (COR) theory.

Textual Explanation of the Visual Addition

We revised the paragraph under “COR Theory and JD-R Model” to reference this addition:

“Although this relationship is not explicitly tested in our statistical models, we now incorporate it visually in the theoretical framework (see Figure 1) to illustrate how emotional regulation ability may influence the choice of emotional labor strategies through resource preservation logic.”

Red Tape in Business Contexts

At the end of the conceptual discussion of red tape, we added the following paragraph:

“While the concept of red tape originates from public administration, recent studies have demonstrated its applicability in business contexts, particularly in large chain retailers where internal procedures, documentation, and surveillance can resemble bureaucratic systems. In contrast, smaller retailers may encounter less formal red tape but still face operational inefficiencies from ambiguous practices or customer-related burdens. These differences highlight the need to further investigate red tape across different types of commercial organizations in future studies.”

2.Refine Hypothesis Logic. The logic chain between H1 and H3—particularly the argument that resource depletion leads to less surface acting—contrasts with traditional models where emotional dissonance leads to more surface acting. It is recommended to strengthen the theoretical rationale for this divergent path and to streamline the language to better highlight the novelty of the authors’ perspective. Additionally, the interpretation that highly customer-oriented employees express authentic negative emotions seems speculative; more literature support or alternative emotion regulation strategies (e.g., cognitive reappraisal) may enhance this explanation.

Response:

We appreciate this thoughtful critique and have strengthened the logic of our hypotheses as follows:

Clarifying the Resource Depletion → Reduced Surface Acting Pathway

We added the following explanation to explicitly distinguish our perspective from traditional dissonance models:

“This logic departs from traditional dissonance-based models, which assume employees always retain enough capacity for surface acting. In high red tape environments, employees may reach a ‘resource exhaustion threshold’ where even minimal effort to feign emotions becomes too costly. Under COR theory, individuals facing continuous resource loss tend to withdraw from optional demands, including superficial displays of compliance. Hence, the decline in surface acting can be interpreted as a strategic disengagement from emotionally taxing behaviors, rather than a failure to comply.”

Incorporating Cognitive Reappraisal for High Customer Orientation Employees

Just before Hypothesis 6, we introduced an alternative emotion regulation explanation:

“However, employees with high customer orientation may not always resort to surface acting when red tape disrupts their service delivery. Instead, they may engage in adaptive emotion regulation strategies such as cognitive reappraisal—reframing frustrating procedures as manageable challenges or temporary obstacles. This reframing helps conserve emotional resources and sustain deep acting or authentic emotional expression, rather than relying on inauthentic surface displays.”

3.Increase Transparency in Methods。Some conditional indirect effects approach non-significance (e.g., in Table 4, the 95% CI for surface acting at high customer orientation includes zero). The authors should advise readers to interpret these results with caution and discuss potential biases due to sample or contextual limitations. Moreover, providing more detail on the localization of survey instruments (e.g., translation process, pilot testing) would strengthen the credibility and cultural relevance of the measurements.

Response:

Thank you for highlighting these important concerns. We have addressed them as follows:

Caution Regarding Confidence Intervals

Following the discussion of Table 4, we added:

“It should be noted that the confidence interval for the conditional indirect effect of red tape on surface acting via negative emotions at high customer orientation includes zero. This suggests that the moderated mediation effect is not statistically robust in this condition, and readers should interpret this result with caution. The lack of significance may be attributed to sample size limitations or contextual variability.”

Survey Localization Process

After the description of the Red Tape scale, we included:

“The survey instruments were first translated from English to Chinese using a forward-backward translation method. Two bilingual experts conducted the initial translation and a separate group of experts reviewed the back-translation for semantic consistency. A pilot study with 35 frontline retail employees was conducted to assess item clarity and contextual relevance, leading to minor adjustments in wording.”

Sample and Generalizability Concerns

In the “Limitations and Future Directions” section, we added:

“The contextual specificity of post-pandemic Beijing retailing may also limit the generalizability of the moderated mediation results. Future studies should expand the sample to cover diverse cities and sectors, and incorporate multi-source data (e.g., managerial ratings or customer satisfaction records) to further validate the observed relationships.”

---

## [Editor Report · Decision Letter 2]

Does the perception of red tape affect the emotional labor of frontline retail staff in China? A post-COVID-19 era

PONE-D-24-49973R2

Dear Dr. Qiang,

We’re pleased to inform you that your manuscript has been judged scientifically suitable for publication and will be formally accepted for publication once it meets all outstanding technical requirements.

Kind regards,

Nicolas Depetris-Chauvin

Academic Editor

PLOS ONE
---

## [Editor Report · Acceptance letter]

PONE-D-24-49973R2

PLOS ONE

Dear Dr. Qiang,

I'm pleased to inform you that your manuscript has been deemed suitable for publication in PLOS ONE. Congratulations! Your manuscript is now being handed over to our production team.

Kind regards,

on behalf of

Dr. Nicolas Depetris-Chauvin

Academic Editor

PLOS ONE